# Adverse pregnancy outcomes are associated with *Plasmodium vivax* malaria in a prospective cohort of women from the Brazilian Amazon

**Jamille Gregório Dombrowski**[1], **André Barateiro**[1], **Erika Paula Machado Peixoto**[1], **André Boler Cláudio da Silva Barros**[2], **Rodrigo Medeiros de Souza**[3], **Taane Gregory Clark**[4], **Susana Campino**[4], **Carsten Wrenger**[1], **Gerhard Wunderlich**[1], **Giuseppe Palmisano**[1], **Sabrina Epiphanio**[5], **Lígia Antunes Gonçalves**[1]☯*, **Claudio Romero Farias Marinho**[1]☯*

**1** Department of Parasitology, Institute of Biomedical Sciences, University of São Paulo, São Paulo, Brazil, **2** Instituto Gulbenkian de Ciência, Oeiras, Portugal, **3** Multidisciplinary Center, Federal University of Acre, Acre, Brazil, **4** Faculty of Infectious & Tropical Diseases, London School of Hygiene & Tropical Medicine, London, United Kingdom, **5** Department of Clinical and Toxicological Analyses, Faculty of Pharmaceutical Sciences, University of São Paulo, São Paulo, Brazil

☯ These authors contributed equally to this work.
* lig.antunes.goncalves@gmail.com (LAG); marinho@usp.br (CRFM)

**Data Availability Statement:** All relevant data are within the manuscript and its Supporting Information files.

## Abstract

### Background

Malaria in Brazil represents one of the highest percentages of Latin America cases, where approximately 84% of infections are attributed to *Plasmodium* (*P.*) *vivax*. Despite the high incidence, many aspects of gestational malaria resulting from *P. vivax* infections remain poorly studied. As such, we aimed to evaluate the consequences of *P. vivax* infections during gestation on the health of mothers and their neonates in an endemic area of the Amazon.

### Methods and findings

We have conducted an observational cohort study in Brazilian Amazon between January 2013 and April 2015. 600 pregnant women were enrolled and followed until delivery. After applying exclusion criteria, 329 mother-child pairs were included in the analysis. Clinical data regarding maternal infection, newborn's anthropometric measures, placental histopathological characteristics, and angiogenic and inflammatory factors were evaluated. The presence of plasma IgG against the *P. vivax* (Pv) MSP1$_{19}$ protein was used as marker of exposure and possible associations with pregnancy outcomes were analyzed. Multivariate logistic regression analysis revealed that *P. vivax* infections during the first trimester of pregnancy are associated with adverse gestational outcomes such as premature birth (adjusted odds ratio [aOR] 8.12, 95% confidence interval [95%CI] 2.69–24.54, $p < 0.0001$) and reduced head circumference (aOR 3.58, 95%CI 1.29–9.97, $p = 0.01$). Histopathology analysis showed marked differences between placentas from *P. vivax*-infected and non-infected

**Funding:** This work was primarily funded by Fundação de Amparo à Pesquisa do Estado de São Paulo-FAPESP (grants no. 2018/20468-0 and 2020/06747-4 to C.R.F.M., 2017/05782-8 and 2020/07251-2 to S.E., and 2015/06106-0 to L.A. G.) (www.fapesp.br); Conselho Nacional de Desenvolvimento Científico e Tecnológico-CNPq (grant no. 302917/2019-5 to C.R.F.M., and 409216/2018-6 to J.G.D) (www.cnpq.br). T.G.C. and S.C. are funded by the Medical Research Council UK (grants no. MR/M01360X/1, MR/ N010469/1, MR/R025576/1, and MR/R020973/1 to T.G.C., and MR/M01360X/1, MR/R025576/1, and MR/R020973/1 to S.C.) (www.mrc.ukri.org); C.W., G.W. and G.P. are funded by FAPESP (grants no. 2015/26722-8 and 2017/03966-4 to C.W., 2017/24267-7 to G.W., and 2018/18257-1, 2018/ 15549-1, and 2020/04923-0 to G.P.). J.G.D and A. B. were supported by fellowships from FAPESP (2016/13465-0 and 2019/12068-5 to J.G.D, and 2017/03939-7 to A.B.); A.B.C.S.B was supported by fellowships from Fundação para a Ciência e a Tecnologia (FCT - PD/BD/138900/2018) (www.fct. pt). The funders had no role in study design, data collection and analysis, decision to publish, or preparation of the manuscript.

**Competing interests:** The authors have declared that no competing interests exist.

pregnant women, especially regarding placental monocytes infiltrate. Placental levels of vasomodulatory factors such as angiopoietin-2 (ANG-2) and complement proteins such as C5a were also altered at delivery. Plasma levels of anti-PvMSP1$_{19}$ IgG in infected pregnant women were shown to be a reliable exposure marker; yet, with no association with improved pregnancy outcomes.

## Conclusions

This study indicates that *P. vivax* malaria during the first trimester of pregnancy represents a higher likelihood of subsequent poor pregnancy outcomes associated with marked placental histologic modification and angiogenic/inflammatory imbalance. Additionally, our findings support the idea that antibodies against PvMSP1$_{19}$ are not protective against poor pregnancy outcomes induced by *P. vivax* infections.

## Author summary

Malaria during pregnancy is associated with adverse effects on the fetus and the newborn. As far as we know, no study has previously investigated in a single work, the link between *Plasmodium vivax* malaria in pregnancy and poor gestational outcomes, alteration of the newborn's anthropometric profile, placental lesions, angiogenic and inflammatory factors, and humoral immunity against the parasite. For this purpose, we investigated the association between *P. vivax* malaria during pregnancy and newborn's anthropometric profile, placental pathology, gestational outcomes, and the presence of IgG against *P. vivax* MSP1$_{19}$ that may confer protection against infection during pregnancy. We performed a large cohort study of malaria during pregnancy that analyzed data from mother-child pairs delivered between 2013 and 2015 in the Southwestern Brazilian Amazonian region. By evaluating data from 329 pregnancies, we found that *P. vivax* malaria during the first pregnancy trimester is significantly associated with the occurrence of preterm birth, low birth weight, and reduced newborn head circumference and body length. We also noted that *P. vivax* malaria in pregnancy promoted placental lesions and homeostasis imbalance, characterized by increased syncytial nuclear aggregates, fibrin deposition, and monocytes/ leukocytes infiltrate, as well as imbalanced angiogenic factors, leptin, and cytokines. We observed that pregnant women with IgG against *P. vivax* MSP1$_{19}$ are not protected against poor pregnancy outcomes caused by *P. vivax* infections during pregnancy. Our observations improve our understanding of the disease and *P. vivax* burden during pregnancy, changing the current paradigm of the outcome of *P. vivax* malaria in pregnancy. That may represent a long-term severe consequence for the affected populations living in *P. vivax*-endemic regions. Our results also indicate that IgG against *P. vivax* MSP1$_{19}$ is not associated with protection from poor pregnancy outcomes, excluding this protein as a possible vaccination target that can prevent adverse outcomes caused by *P. vivax* infections during pregnancy.

## Introduction

Pregnant women are highly susceptible to malaria and its deleterious effects. Recently, the World Health Organization Malaria Report pointed the often-neglected malaria cases in

pregnancy (MiP) that occur mainly in Africa, where around 11 million pregnant women were estimated to have developed the disease during 2018 [1]. Africa is characterized by a high and stable transmission of *Plasmodium* (*P.*) *falciparum*. Despite being frequently under the spotlight of malaria control campaigns, infections during pregnancy are still a burden associated with maternal anemia, abortion, premature delivery, and low birth weight (LBW) [2–7]. Few studies have focused on the burden of MiP in low transmission regions like Latin America, where *P. vivax* is the most prevalent species [1]. Its infections are often considered benign, a paradigm that is frequently supported by inconsistent reports that tend to focus on *P. falciparum* malaria episodes during pregnancy; this makes the evaluation of the real impact of *P. vivax* monoinfections somehow challenging [8–11].

In *P. falciparum* MiP, poor pregnancy outcomes are mainly linked to severe placental inflammation triggered by the sequestration of infected erythrocytes (which characterize placental malaria, PM) [12]. This process results in dramatic histological changes of the placental structure [11,13], which ultimately dysregulate local homeostasis and contribute to poor pregnancy outcomes [2]. This phenomenon is not yet clearly elucidated in *P. vivax* MiP mainly due to scarce and controversial evidence concerning: (1) placental adhesion [14,15]; (2) severe placental histopathology [9–11]; and (3) subsequent poor gestational outcomes [16–18]. The harmful effects of *P. vivax* infections are most likely a consequence of systemic inflammation and production of cytokines [19], such as TNF-α and IFN-γ, which are linked to disease severity, or IL-10 that is associated with better disease prognosis [20]. Accordingly, *P. vivax*-induced inflammation during pregnancy may be implicated in placental structural abnormalities and leukocytes infiltration, as well as in angiogenic imbalance, which altogether can directly or indirectly impact fetal development [11,13,21]. Nevertheless, to our knowledge, studies addressing these parameters together are still missing.

Attenuation of clinical symptoms frequently occurs after naturally acquiring immunity due to frequent exposures to *Plasmodium* spp. In *P. falciparum* MiP, loss of protection is mainly attributed to lack of immunity against the *P. falciparum* erythrocyte membrane protein 1 (PfEMP-1) variant VAR2CSA that binds to chondroitin sulfate A (CSA) in the placenta. In subsequent pregnancies, the development of PfEMP1$_{var2csa}$ specific antibodies is associated with protection from MiP [22–26]; yet, in *P. vivax* MiP, the establishment of a clear association between pre-existing acquired immunity and improved pregnancy outcomes is yet to be achieved.

Herein, we report the deleterious relationship between *P. vivax* monoinfections, poor pregnancy outcomes, placental histopathology, cytokine and angiogenic imbalance, and humoral immune response in pregnant women from the Brazilian Amazon. Our results suggest that *P. vivax* infections during the first gestational trimester are associated with poor pregnancy outcomes. The presence of IgG against *P. vivax* merozoite surface protein 1 (*P. vivax* MSP1$_{19}$) is not associated with protection against the deleterious effects of *P. vivax* MiP.

## Methods

### Ethics statement

This study was approved by the Ethics Committee from the University of São Paulo (Plataforma Brasil, CAAE: 03930812.8.0000.5467 and 90474318.4.0000.5467), according to resolution n˚ 466/12 of the Brazilian National Health Committee and in compliance with the Declaration of Helsinki. All subjects or their legal guardians (if minors) gave written informed consent. This study followed a specified analysis plan (**S1 Text**) and the Strengthening the Reporting of Observational Studies in Epidemiology (STROBE) statement guidelines (**S1 STROBE Checklist**).

### Setting and study design

We have conducted an observational cohort study in five municipalities (Cruzeiro do Sul, Mâncio Lima, Rodrigues Alves, Marechal Thaumaturgo, and Porto Walter) in the Juruá Valley, located in the Brazilian Amazon southwest region (Acre State, Brazil). This region is considered highly endemic for malaria, with an annual parasite incidence (API) above 100 cases per 1,000 inhabitants. Juruá Valley is also characterized by a significant prevalence of *P. vivax* infections, responsible for 70–80% of total malaria cases [27,28]. A complete region characterization with detailed procedures and methods used for data and sample collection has been described elsewhere [11,13,21]. Briefly, 600 pregnant women were recruited during their first antenatal care (ANC) visit. These women were followed until delivery, between January 2013 and April 2015. Data was collected on socioeconomic, clinical, and obstetric variables at the time of recruitment, together with peripheral blood samples. In parallel, thick and thin peripheral blood smears were performed to screen for malaria parasites by microscopy, which were later confirmed by PCR. Each pregnant woman was followed by a trained nurse, which involved at least two domiciliary visits, at the second and third trimester, to monitor their clinical state and collect a peripheral blood sample, in addition to the usual prenatal care visit to the health care services. An additional blood sample was collected in each malaria episode during pregnancy. At the time of delivery, clinical data were collected from mother and newborn and performed a placental biopsy and blood sampling. Only women with *P. vivax* monoinfections and their non-infected counterparts were considered for this study (**Fig 1A**). We evaluated mother-child pairs at delivery in the Women and Children's Hospital of Juruá Valley located in Cruzeiro do Sul.

### Malaria screening and treatment

Malaria during pregnancy was first diagnosed from thin and thick blood smears by microscopy analysis, which was performed by the endemic surveillance team of Juruá Valley. Upon diagnosis, all women infected with *P. vivax* during pregnancy were treated with chloroquine (25mg/kg of body weight) in divided doses over a 3-day period. Successful treatment was confirmed subsequently. No hypnozontocidal treatment with primaquine was performed, following the Brazilian Ministry of Health (MoH) guidelines regarding medical prescription in pregnant women [29]. Of note, the Intermittent Preventive Treatment in pregnancy (IPTp) is not employed in Brazil. All samples collected throughout pregnancy (recruitment, malaria episodes, and domiciliary visits) and at delivery were further screened by molecular diagnosis using the real-time PCR technique (PET-PCR) as described elsewhere [13,30].

### Exclusion criteria

Pregnant women were excluded from further analysis if they self-informed any alcohol consumption, smoking, and/or if presenting any other diagnosed infectious diseases (e.g. Toxoplasmosis, Rubella, Cytomegalovirus and Herpes simplex (TORCH), HIV, Hepatitis B and C virus, Syphilis), and/or other comorbidities (e.g. hypertension, pre-eclampsia/eclampsia, diabetes mellitus, and newborn with congenital malformation). Likewise, pregnant women infected with other *Plasmodium* species (*P. falciparum* or mixed infections) were excluded after confirmation by PET-PCR (**Fig 1A**).

### Laboratory procedures

Peripheral and placental blood (from the maternal side) was collected aseptically in heparinized tubes and separated into plasma and whole blood cells after centrifugation. All samples

**A**

600 Mother-child pairs
[Delivered 01/2013 - 04/2015]

86 Mother-child pairs excluded
- 52 Alcohol and smoke
- 30 Other infectious diseases
- 1 Pre-eclampsia
- 2 >1 fetus
- 1 Congenital anomaly

514 Mother-child pairs
initially included

39 Mother-child pairs excluded
Missing data

146 Mother-child pairs excluded
- 72 Mixed-infected
- 74 *P. falciparum*-infected

329 Mother-child pairs included in the analysis
- 170 Non-infected mothers
- 159 *P. vivax*-infected mothers

**B**

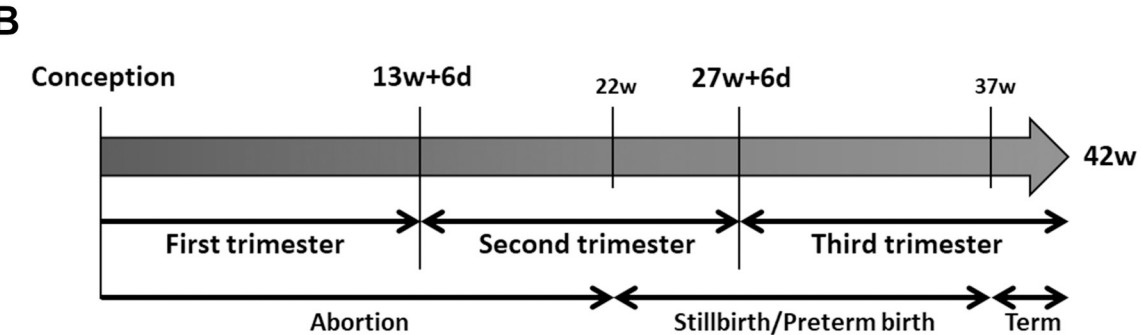

**Fig 1. Description of study participants and definitions.** (A) Flow chart detailing exclusion criteria applied prior data analysis; (B) Definitions of pregnancy partitions used in the report. Mixed infection–*P. vivax* and *P. falciparum* infections occurring at the same or different times during pregnancy; w—weeks; d—days.

were stored at -80˚C. Total DNA was obtained from whole blood cells using a commercially available extraction kit (QIAmp DNA Mini Kit, Qiagen), following the manufacturer's instructions. Fragments of placental tissue were collected and stored in 10% neutral buffered formalin at 4˚C, until processing. Placental tissue was sectioned and embedded in paraffin using standard procedures; afterward, it was stained with Hematoxylin-Eosin (H&E) or Giemsa for histological examination. All histological evaluation methods were optimized by our group and are described elsewhere [11,13]. Placental images were captured using a Zeiss Axio Imager M2 light microscope equipped with a Zeiss Axio Cam HRc camera. Placental parameters were evaluated and analyzed using Image J software (http://imagej.nih.gov/ij). All measurements were performed by two investigators blinded to group and outcome. Cases that proved to be contradictory between observers were re-evaluated until consensus was reached.

## Placental weight and newborn's anthropometric measurements

Placental weight was measured (grams (g)) immediately after expulsion/extraction using a digital semi-analytical scale (Shimadzu), with a capacity of 3.2 kilograms (kg) and a precision of 0.01 g. Umbilical cord, membranes, and fetal blood contained within the large chorionic vessels and intervillous space, were included in weight measurement. Newborns' anthropometric measures were obtained after birth (maximum within a 24-hour period) by trained obstetrician nurses. Weight was measured (g) using digital pediatric scales (Toledo), with a maximum capacity of 15 kg and a precision of 5 g. Length, head and chest circumferences were measured using a non-stretching flexible measuring tape and expressed as centimeters (cm). Rohrer's ponderal index was calculated as the newborn's weight (g) divided by the cube of the length (cm).

## Gestational age estimation and other definitions

Gestational age (GA) was assessed during the first trimester of pregnancy using crown-rump length measured by trans-abdominal ultrasound. In the absence of this examination within this period, the GA was estimated by the woman's last menstrual period (LMP) together with the earliest ultrasound performed. The LMP method is recommended by the Brazilian MoH for GA calculation. Low birth weight (LBW) was defined as birth weight below 2500 g. Preterm birth was defined as birth before the 37th week of gestation, abortion as birth occurring before the 22nd week of gestation, and stillbirth as fetal death occurring between the 22nd week of gestation and time of delivery (**Fig 1B**). Term LBW was defined as term birth (after 37th week of gestation) with weight below 2500 g. Small for gestational age (SGA), reduced head circumference, and reduced body length were defined as: birth weight, head circumference and length below the 10th percentile sex-specific for GA according to the INTERGROWTH-21st standards [31]. Maternal anemia was defined as hemoglobin levels lower than 11 g/dL. Placental malaria was diagnosed according to a previously established classification [12].

## Angiogenic factors and leptin measurement

Angiogenic factors (angiopoietins 1 and 2 (ANG-1 and ANG-2), soluble TEK receptor tyrosine kinase (sTIE-2), vascular endothelial growth factor A (VEGF-A), receptors fms-like tyrosine kinase-1 (sFlt-1) and soluble vascular endothelial growth factor receptor 2 (sVEGFR-2)), and the leptin hormone were measured in placental plasma (1:20 dilution for all factors) using the DuoSet ELISA development kits (R&D), according to manufacturer's guidelines.

## Cytokines/anaphylatoxins measurement by cytometric bead array (CBA)

Interleukin (IL-) 1β, IL-6, IL-8, IL10, IL-12p70, and TNF-α were detected and quantified in placental plasma by CBA Human Inflammatory kit (BD Biosciences), according to the manufacturer's protocol. Complement (C3a, C4a, and C5a) activation was evaluated in placental plasma through CBA Human Anaphylatoxin kit (BD Biosciences). Samples were analyzed in a two-laser BD FACSCalibur flow cytometer with CellQuest version 5.2 software (BD Biosciences) and concentrations computed using FCAP array software version 3.0.1 (BD Biosciences).

## Measurement of total IgG antibodies against *Plasmodium vivax* merozoite surface protein 1 (PvMSP1$_{19}$)

For the study of total IgG antibodies against PvMSP1$_{19}$, samples collected at the first time-point (recruitment) were used; yet, we excluded infected pregnant women who were recruited after having already one or more malaria episodes during that pregnancy. Enzyme-linked immunosorbent assay (ELISA) was performed using high-binding 96-well microplates (Costar, Cambridge, MA) coated with Glutathione S-transferase (GST)-PvMSP1$_{19}$ (31.25 ng/well), produced as described elsewhere [32,33]. In parallel, GST was used as a control for non-specific binding. The optimal concentrations were determined by plasma serial titration of positive and negative controls. Plates were incubated at 4˚C overnight and washed four times with phosphate buffered saline with 0.05% Tween 20 (PBS-T). PBS-T containing skimmed milk was used for blocking (4%) and diluting plasma and antibodies (1%). Each plasma sample (50 μL/well) was diluted 1:200. After a 1-hour incubation at room temperature, antibody-antigen complexes were detected with HRP-conjugated mouse anti-human IgG (IG266) (Novus, Cat# NBP2-34648H) at a 1:3,000 dilution. Enzyme-substrate reaction was revealed with tetramethylbenzidine (TMB)/H$_2$O$_2$ at room temperature (dark conditions). After 10 minutes, the reaction was interrupted with 2N H$_2$SO$_4$. Absorbance values were measured at 450/595 nm using a CLARIOstar Plus plate reader (BMG Labtech). Corrected absorbance values were obtained by subtracting absorbance readings with GST ran on the same microplate. Absorbance data used in statistical analyses correspond to corrected values. Reactivity indices (RIs) were calculated as the ratio between each test sample's corrected absorbance values and a cut-off value for antigen, corresponding to the average corrected absorbance for samples from 10 malaria-naïve blood donors plus 3 standard deviations. Positive samples were considered as having RIs greater than 1.

## Statistical analysis

Data were analyzed using Stata/SE 14.2 (Stata, College Station, TX, USA) and GraphPad Prism 6.0 software (GraphPad Software Inc., San Diego, CA, USA). Continuous variables were summarized using mean and standard deviation (SD) values as well as median values and interquartile ranges (IQRs). Categorical variables were summarized using frequencies and percentages. Differences between groups were evaluated using the nonparametric Kruskal-Wallis test followed by the Dunn's post hoc multiple comparison test and the Mann-Whitney test as appropriate. Differences between categorical data and proportions were analyzed using Pearson′s chi-squared test or Fisher's exact test depending on the type of variables. All p values were 2-sided and interpreted at a significance level of 0.05. To compare the association between malaria and other variables (clinical characteristics, newborns' anthropometric measurements, placental parameters, angiogenic and inflammatory factors, and anti-PvMSP1$_{19}$ IgG), we used linear regression, adjusting for maternal age, number of gestations, residence,

education level, and occupation. To assess associations between malaria and adverse pregnancy outcomes, the presence of anti-PvMSP1$_{19}$ IgG and outcomes in the context of MiP, adjusted odds ratios (aOR) with 95% confidence intervals (CI) were estimated using multiple logistic regression, which included infection by malaria (no / yes), maternal age ($\geq$ 18 years old / $\leq$ 17 years old), number of gestations (two or more / one), residence (urban / rural), education level ($\geq$ secondary / $\leq$ primary), occupation (others / housewife) and presence of anti-PvMSP1$_{19}$ IgG (no / yes) as explanatory variables and adverse pregnancy outcomes (yes / no) as response variables. The first category for each explanatory variable was considered as the reference.

## Results

### Study population and baseline characteristics

From 600 pregnant women enrolled in the major study, 329 (54.8%) were eligible for further analysis after applying the exclusion criteria and malaria parasite species diagnosis by PET-PCR (**Fig 1A**). Of these, 170 (51.7%) were classified as non-infected and 159 (48.3%) as *P. vivax* (*Pv*)-infected during gestation (**Fig 1A**). The overall socio-demographic profile of the stratified participants is depicted in **Table 1**. Group stratification was performed by the trimester in which the first infection occurred, considering that time of infection is known to be associated with MiP poor pregnancy outcomes [34]. **Fig 1B** illustrates a standard gestational window with the hypothetical poor outcomes in unsuccessful pregnancies. **Table 1** shows some heterogeneity between the groups since pregnant women in the infected groups tend to be younger, with reduced educational level and rural residence. Occupation also varies significantly between the groups, but no difference was found regarding the number of previous pregnancies. The aforementioned characteristics were used as explanatory variables in the following multiple logistic and linear regressions analysis due to reported heterogeneity and their possible impact as confounding effects.

### *P. vivax* infections during the first trimester are associated with MiP poor pregnancy outcomes

Malaria in pregnancy is known to impact proper fetal development [2]. We assessed pregnancy outcomes and newborns' anthropometric measurements to ascertain if *P. vivax* infections at a given gestational trimester are associated with impaired fetal development and poor outcomes. Multiple logistic regression analysis revealed that *P. vivax* infections during the first trimester of pregnancy increase the odds of premature birth (aOR 8.12, 95%CI 2.69–24.54, $p < 0.0001$), LBW (aOR 4.34, 95%CI 1.49–12.62, $p = 0.007$), as well as LBW in those born at term (aOR 4.75, 95%CI 1.06–21.35, $p = 0.04$). First trimester infections were also associated with reduced head circumference (aOR 3.58, 95%CI 1.29–9.97, $p = 0.01$) and newborn's length (aOR 4.48, 95%CI 1.66–12.06, $p = 0.003$) (**Fig 2**). No differences were observed for the overall birth weight and length (**Fig 3A and 3D** and **S1 Table**). Though, in line with our findings, we observed statistically significant differences in head and chest circumference of children born from women infected with *P. vivax* during the first trimester and those born from non-infected women (median [IQR], head–*Pv*-1$^{st}$ tri 34.0 cm [32.0–34.0] vs. NI 34.0 cm [33.0–35.0], $p = 0.007$; chest–*Pv*-1$^{st}$ tri 33.0 cm [32.0–34.0] vs NI 34.0 cm [33.0–35.0], $p = 0.006$) (**Fig 3B and 3C** and **S1 Table**). Our results suggest that *P. vivax* infections during the first gestational trimester promote preterm birth and impair fetal development, possibly due to clinical manifestations associated with MiP.

**Table 1. Demographic characteristics of the mothers participating in the study, according to infection and gestational trimester of first infection.**

| Characteristics | Non-Infected (N = 170) | *P. vivax* (N = 159) | *p*-value[a] | *P. vivax* - 1st tri (N = 52) | *p*-value[b] | *P. vivax* - 2nd tri (N = 54) | *p*-value[c] | *P. vivax* - 3rd tri (N = 53) | *p*-value[d] |
|---|---|---|---|---|---|---|---|---|---|
| Age, years, median (IQR) | 23.0 (19.0–28.0) | 21.0 (17.0–26.0) | 0.002 | 22.5 (18.0–25.0) | 0.12 | 19.0 (17.0–26.0) | 0.002 | 21.0 (17.0–26.0) | 0.01 |
| Gravidity, median (IQR) | 2.0 (1.0–3.0) | 2.0 (1.0–3.0) | 0.38 | 2.0 (1.0–3.0) | 0.13 | 1.0 (1.0–3.0) | 0.31 | 2.0 (1.0–3.0) | 0.11 |
| Education level, n (%) | | | < 0.0001 | | 0.001 | | 0.004 | | < 0.0001 |
| No education | 0 | 3 (1.9) | | 0 | | 0 | | 3 (5.7) | |
| Primary | 7 (4.1) | 17 (10.7) | | 4 (7.7) | | 4 (7.4) | | 9 (17.0) | |
| ≥ Secondary | 163 (95.9) | 139 (87.4) | | 48 (92.3) | | 50 (92.6) | | 41 (77.3) | |
| Occupation, n (%) [e] | | | 0.001 | | 0.18 | | 0.03 | | < 0.0001 |
| Farmer | 3 (1.8) | 9 (5.7) | | 4 (7.9) | | 1 (1.8) | | 4 (7.5) | |
| Housewife | 77 (45.3) | 86 (54.1) | | 22 (43.1) | | 30 (55.6) | | 34 (64.2) | |
| Student | 36 (21.2) | 41 (25.8) | | 12 (23.5) | | 16 (29.6) | | 13 (24.5) | |
| Other occupation | 54 (31.7) | 22 (13.8) | | 13 (25.5) | | 7 (13.0) | | 2 (3.8) | |
| Rural residence, n (%) | 8 (4.7) | 44 (27.7) | < 0.0001 | 12 (23.1) | < 0.0001 | 16 (29.6) | < 0.0001 | 16 (30.2) | < 0.0001 |

Abbreviations: N, total number of individuals; tri, trimester; NA, not applicable. Results are presented as median and interquartile range (IQR) or total number of events (n) and percentage (%). Statistical tests were applied according to the type of variable (Mann-Whitney, Chi-square or Kruskal-Wallis with Dunn's corrections).

[a] Differences between Non-Infected and *P. vivax* group.

[b] Differences between Non-Infected and *P. vivax* infection in the 1st trimester.

[c] Differences between Non-Infected and *P. vivax* infection in the 2nd trimester.

[d] Differences between Non-Infected and *P. vivax* infection in the 3rd trimester.

[e] Occupation was recorded in 158 *P. vivax*-infected pregnant.

## Clinical and infection characteristics of pregnant women with *P. vivax* MiP

As for the aforementioned outcomes, pregnant women infected with *P. vivax* had some clinical characteristics that are worth mentioning (**Table 2**). Women infected during the first trimester had reduced gestational age at delivery when compared with non-infected counterparts (median [IQR], *Pv*-1st tri 39.0 weeks [37.0–40.0] vs. NI 40.0 weeks [39.0–40.0], *p* = 0.006). These results are in line with our observations that suggest an association between first trimester infections and pregnancy outcomes such as LBW and preterm delivery (**Fig 2** and **Table 2**). Reduced maternal weight gain was also observed in this group (median [IQR], *Pv*-1st tri 10.0 Kg [7.0–12.0] vs. NI 13.5 Kg [10.0–16.8], *p* = 0.004). Anemia is frequently found in pregnant women infected with *P. vivax*, which has been suggested to be one of the leading causes of impaired fetal development during *P. vivax* MiP [16,35]. However, a statistically significant reduction in hematocrit was only observed for the overall *Pv*-infected group (median [IQR], *Pv* 34.8% [32.4–37.5] vs. NI 36.6% [34.5–38.6], *p* = 0.008) and increased clinical anemia in those infected during the third trimester (*Pv*-3rd tri 28.3% vs. NI 14.1%, *p* = 0.02). Additionally, women in the *Pv*-infected group reported more malaria episodes before current pregnancy than their non-infected counterparts regardless the infection trimester (**Table 2**). Pregnant women infected during the first trimester had higher chances of having three or more infections than infected women from other groups. Parasitemia tend to be similar for the first infection across gestational trimesters; yet, it seems to decrease on average from the second infection onward (**Table 3**). 8.1% of women infected in the first trimester had past PM, while 4.6 and 4.3% of women infected during the second and third trimesters had active acute

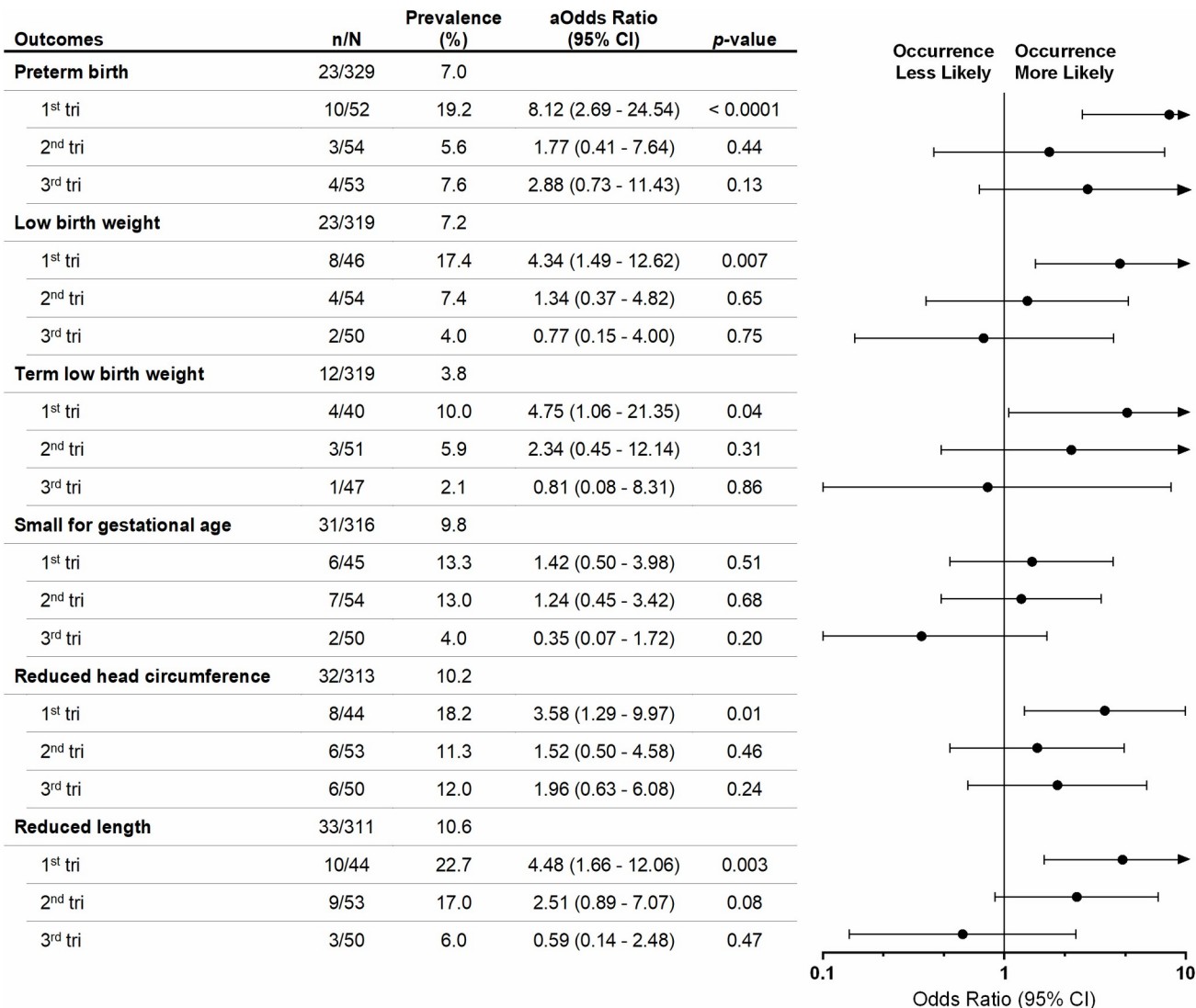

| Outcomes | n/N | Prevalence (%) | aOdds Ratio (95% CI) | p-value |
|---|---|---|---|---|
| **Preterm birth** | 23/329 | 7.0 | | |
| 1st tri | 10/52 | 19.2 | 8.12 (2.69 - 24.54) | < 0.0001 |
| 2nd tri | 3/54 | 5.6 | 1.77 (0.41 - 7.64) | 0.44 |
| 3rd tri | 4/53 | 7.6 | 2.88 (0.73 - 11.43) | 0.13 |
| **Low birth weight** | 23/319 | 7.2 | | |
| 1st tri | 8/46 | 17.4 | 4.34 (1.49 - 12.62) | 0.007 |
| 2nd tri | 4/54 | 7.4 | 1.34 (0.37 - 4.82) | 0.65 |
| 3rd tri | 2/50 | 4.0 | 0.77 (0.15 - 4.00) | 0.75 |
| **Term low birth weight** | 12/319 | 3.8 | | |
| 1st tri | 4/40 | 10.0 | 4.75 (1.06 - 21.35) | 0.04 |
| 2nd tri | 3/51 | 5.9 | 2.34 (0.45 - 12.14) | 0.31 |
| 3rd tri | 1/47 | 2.1 | 0.81 (0.08 - 8.31) | 0.86 |
| **Small for gestational age** | 31/316 | 9.8 | | |
| 1st tri | 6/45 | 13.3 | 1.42 (0.50 - 3.98) | 0.51 |
| 2nd tri | 7/54 | 13.0 | 1.24 (0.45 - 3.42) | 0.68 |
| 3rd tri | 2/50 | 4.0 | 0.35 (0.07 - 1.72) | 0.20 |
| **Reduced head circumference** | 32/313 | 10.2 | | |
| 1st tri | 8/44 | 18.2 | 3.58 (1.29 - 9.97) | 0.01 |
| 2nd tri | 6/53 | 11.3 | 1.52 (0.50 - 4.58) | 0.46 |
| 3rd tri | 6/50 | 12.0 | 1.96 (0.63 - 6.08) | 0.24 |
| **Reduced length** | 33/311 | 10.6 | | |
| 1st tri | 10/44 | 22.7 | 4.48 (1.66 - 12.06) | 0.003 |
| 2nd tri | 9/53 | 17.0 | 2.51 (0.89 - 7.07) | 0.08 |
| 3rd tri | 3/50 | 6.0 | 0.59 (0.14 - 2.48) | 0.47 |

**Fig 2. Association between adverse pregnancy outcomes and *P. vivax* infections according to the trimester of first infection.** Forest plot of odds ratios of adverse events in newborns from women infected during pregnancy compared with newborns from non-infected women is shown. n/N—number of events by total number of individuals in each group; aOdds Ratio—adjusted Odds Ratio; CI—confidence interval; tri—gestational trimester. Odds ratios were adjusted (aOR) for maternal age, gravidity status, residence, education level and occupation. Preterm birth–birth < 37th week of gestation; Low birth weight–birth weight < 2500 g; Term low birth weight–birth weight < 2500 g from 37th week of gestation; Small for gestational age, Reduced head circumference and Reduced body length–birth weight, head circumference and length (respectively) < 10th centile for sex-specific gestational age. *P-values* were estimated through multiple logistic regression methods.

PM, respectively. This was established by the presence of hemozoin in milder or moderate amounts. However, it was absent in more than 90% of the examined placentas, regardless of the infection's gestational trimester. Our observations support the idea that systemic effects rather than PM are linked to the outcomes observed during *P. vivax* MiP in our cohort.

## *P. vivax* infections across gestational trimesters lead to diverse placental histologic alterations and angiogenic and cytokine imbalance

Reduced PM prevalence is frequently observed in *P. vivax* infections because, contrary to *P. falciparum* infections, *P. vivax* sequestration in the placenta is controversial [14,15] and studies

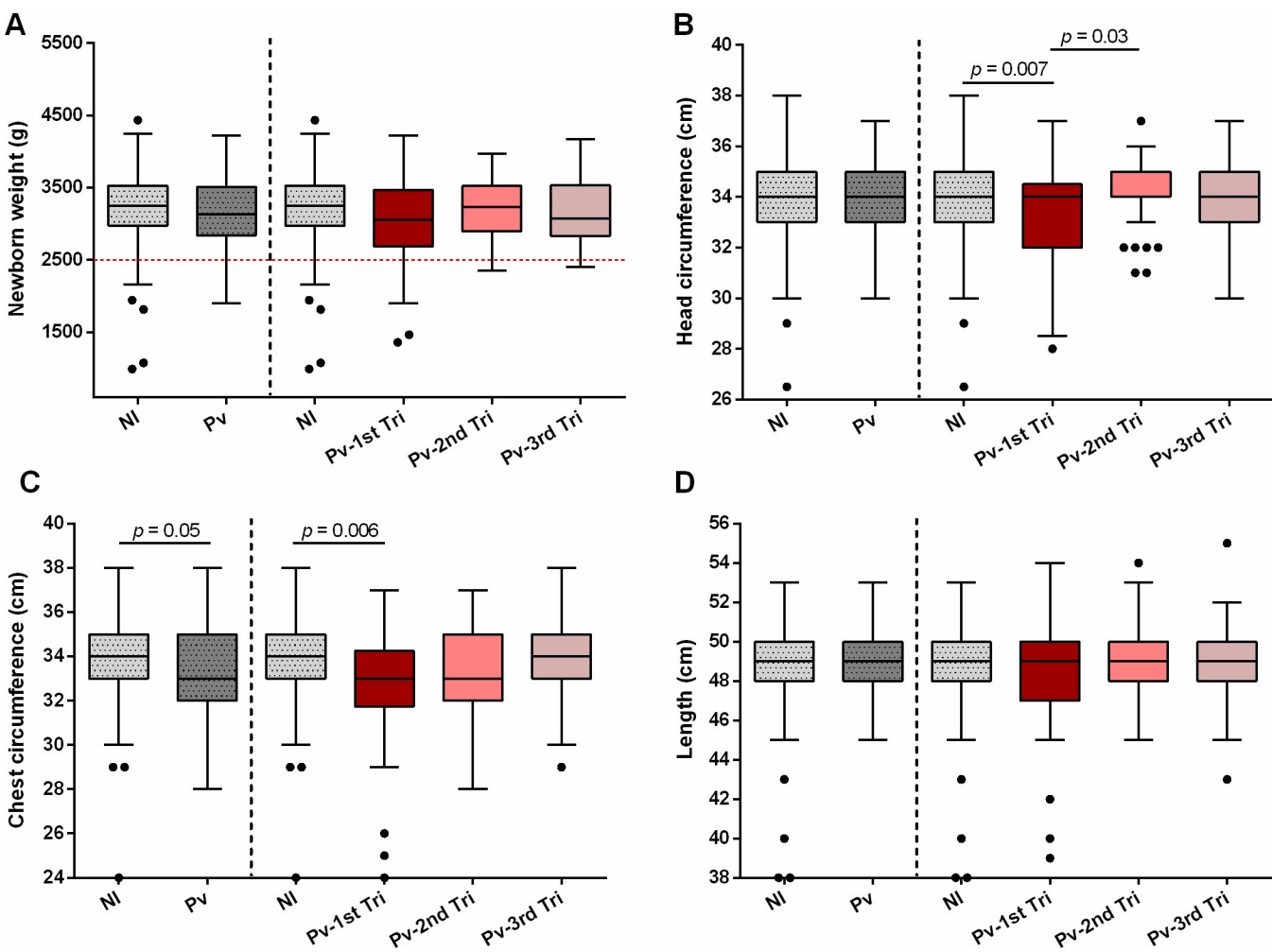

**Fig 3. Anthropometric measurements of newborns according to maternal infection and trimester of first infection.** (A) Newborn weight; (B) Head circumference; (C) Chest circumference; and, (D) Length. NI–non-infected (N = 160–169); Pv–*P. vivax*-infected pregnant women (N = 130–150); Pv-1st tri–*P. vivax* infection 1st trimester (N = 38–46); Pv-2nd tri–*P. vivax* infection 2nd trimester (N = 43–54); Pv-3rd tri–*P. vivax* infection 3rd trimester (N = 49–50). Weight is given in grams (g). Head and chest circumferences, and body length are informed in centimeters (cm). Data are represented as Tukey boxplots, with the bottom and top of each box representing the first and third quartiles; the line inside the box is the median, and the whiskers represent the lowest and the highest data within 1.5 IQR of the first and upper quartiles; and the circles, outliers. Differences between groups were determined by multiple liner regression (adjusted for maternal age, gravidity status, residence, education level, and occupation).

that report this event frequently have reduced sample sizes. Nevertheless, placental histologic changes had been observed [9–11]. To determine whether *Pv*-infections across the gestational trimesters result in considerable histologic and cytological alterations, we have evaluated several histologic parameters in more than 100 placentas from *Pv*-infected women (**Fig 4 and S2 Table**). We observed that placentas from *Pv*-infected women had a significant increase in syncytial nuclear aggregates (SNA), fibrin deposition, monocytes and leukocytes infiltrate (median [IQR], SNA–*Pv* 15.0 [11.0–21.0] vs. NI 13.0 [10.0–17.0], $p = 0.001$; fibrin–*Pv* 2.5 [1.9–2.8] vs. NI 1.9 [1.9–2.8], $p = 0.001$; monocytes–*Pv* 8.0 [6.0–13.0] vs NI 4.0 [2.0–6.0], $p < 0.0001$; leucocytes–*Pv* 21.5 [14.0–29.0] vs. NI 15.0 [9.0–21.0], $p = 0.004$) (**Fig 4 and S2 Table**). Differences between placentas from non-infected women and those from women infected in the first trimester were observed regarding SNA and monocyte infiltrate (median [IQR], SNA–*Pv*-1st tri 19.0 [11.0–22.0] vs. NI 13.0 [10.0–17.0], $p = 0.02$; monocytes–*Pv*-1st tri 7.0 [5.0–9.5] vs NI 4.0 [2.0–6.0], $p = 0.01$) (**Fig 4A and 4C and S2 Table**). Interestingly,

**Table 2. Clinical characteristics of the mothers participating in the study, according to infection and gestational trimester of first infection.**

| Characteristics | Non-Infected (N = 170) | *P. vivax* (N = 159) | p-value[a] | *P. vivax* - 1st tri (N = 52) | p-value[b] | *P. vivax* - 2nd tri (N = 54) | p-value[c] | *P. vivax* - 3rd tri (N = 53) | p-value[d] |
|---|---|---|---|---|---|---|---|---|---|
| **Clinical outcomes** | | | | | | | | | |
| Gestational age at delivery, weeks, median (IQR) [e] | 40.0 (39.0–40.0) | 39.0 (38.0–40.0) | 0.09 | 39.0 (37.0–40.0) | 0.006 | 39.0 (39.0–40.0) | 0.25 | 40.0 (38.0–40.0) | 0.33 |
| Maternal weight gain, Kg, median (IQR) [f] | 13.5 (10.0–16.8) | 10.8 (8.0–14.0) | < 0.0001 | 10.0 (7.0–12.0) | 0.004 | 11.0 (7.5–15.0) | 0.10 | 10.8 (8.0–15.0) | 0.26 |
| Hematocrit during pregnancy, % [g] | 36.6 (34.5–38.6) | 34.8 (32.4–37.5) | 0.008 | 34.9 (32.7–37.8) | 0.37 | 34.5 (32.4–36.9) | 0.11 | 34.9 (32.4–37.5) | 0.24 |
| Hematocrit at delivery, % [h] | 36.2 (33.6–39.1) | 35.4 (32.4–37.6) | 0.70 | 36.1 (33.0–39.1) | 0.93 | 35.3 (32.8–37.3) | 1.00 | 34.2 (30.7–37.4) | 0.33 |
| Hemoglobin during pregnancy, g/dL, median (IQR) [i] | 12.2 (11.6–12.8) | 11.7 (10.8–12.3) | 0.16 | 12.0 (10.9–12.4) | 0.67 | 11.5 (10.7–12.1) | 0.54 | 11.6 (10.8–12.6) | 0.96 |
| Hemoglobin at delivery, g/dL, median (IQR) [j] | 11.8 (11.2–12.8) | 11.6 (10.7–12.5) | 0.64 | 12.1 (10.8–12.6) | 0.97 | 11.6 (11.0–12.3) | 1.00 | 11.3 (10.1–12.3) | 0.32 |
| Anemia, n (%) [k] | 24 (14.1) | 32 (20.1) | 0.15 | 10 (19.2) | 0.38 | 7 (13.0) | 1.00 | 15 (28.3) | 0.02 |
| Antenatal care visits, median (IQR) [l] | 8.0 (6.0–10.0) | 6.0 (5.0–8.0) | < 0.0001 | 6.0 (4.0–9.0) | 0.07 | 6.0 (5.0–8.0) | 0.004 | 6.0 (5.0–8.0) | 0.003 |
| Placental weight, g, median (IQR) [m] | 577.4 (508.3–658.9) | 541.8 (482.7–613.3) | 0.12 | 521.4 (455.0–605.3) | 0.06 | 554.6 (500.9–641.5) | 0.89 | 541.4 (497.8–614.5) | 1.00 |
| **Gestational outcomes, n (%)[n]** | | | | | | | | | |
| Abortion | 1 (0.6) | 6 (3.8) | 0.06 | 6 (11.5) | 0.001 | 0 | 1.00 | NA | - |
| Stillbirth | 2 (1.2) | 1 (0.6) | 1.00 | NA | - | 0 | 1.00 | 1 (1.9) | 0.56 |
| Preterm birth | 6 (3.5) | 17 (10.7) | 0.02 | 10 (19.2) | 0.001 | 3 (5.6) | 0.45 | 4 (7.6) | 0.25 |
| Low birth weight | 9 (5.3) | 14 (9.3) | 0.20 | 8 (17.4) | 0.01 | 4 (7.4) | 0.52 | 2 (4.0) | 1.00 |
| Term low birth weight | 4 (2.4) | 8 (5.0) | 0.25 | 4 (7.7) | 0.09 | 3 (5.6) | 0.36 | 1 (1.9) | 1.00 |
| Small for gestational age | 16 (9.6) | 15 (10.1) | 1.00 | 6 (13.3) | 0.42 | 7 (13.0) | 0.45 | 2 (4.0) | 0.26 |
| **Malaria history, n (%) [o]** | | | | | | | | | |
| Previous episodes before current gestation | 92 (54.1) | 139 (87.4) | < 0.0001 | 47 (90.4) | < 0.0001 | 47 (87.0) | < 0.0001 | 45 (86.5) | < 0.0001 |
| Previous episodes in other gestations | 7 (7.8) | 25 (28.4) | 0.0004 | 7 (21.2) | 0.05 | 7 (26.9) | 0.01 | 11 (36.7) | 0.0004 |

Abbreviations: N, total number of individuals; tri, trimester; Kg, kilograms; g, grams; NA, not applicable. Results are presented as median and interquartile range (IQR) or total number of events (n) and percentage (%). Statistical tests were applied according to the type of variable (Mann-Whitney, Chi-square or Kruskal-Wallis with Dunn's corrections or Multiple linear regression, adjusted for maternal age, gravidity, residence, education, and occupation).

[a] Differences between Non-Infected and *P. vivax* group.

[b] Differences between Non-Infected and *P. vivax* infection in the 1st trimester.

[c] Differences between Non-Infected and *P. vivax* infection in the 2nd trimester.

[d] Differences between Non-Infected and *P. vivax* infection in the 3rd trimester.

[e] Gestational age was recorded in 168 non-infected and 157 *P. vivax* infected pregnant women.

[f] Maternal weight gain was recorded in 163 non-infected and 126 *P. vivax* infected pregnant women. It was determined as the final weight minus the initial weight during pregnancy.

[g] Hematocrit during pregnancy was recorded in 152 non-infected and 136 *P. vivax* infected pregnant women.

[h] Hematocrit at delivery was recorded in 116 non-infected and 101 *P. vivax* infected pregnant women.

[i] Hemoglobin during pregnancy was recorded in 157 non-infected and 136 *P. vivax* infected pregnant women.

[j] Hemoglobin at delivery was recorded in 116 non-infected and 100 *P. vivax* infected pregnant women.

[k] Defined as hemoglobin <11 g/dL.

[l] Antenatal care visits were recorded in 164 non-infected and 139 *P. vivax* infected pregnant women.

[m] Placental weight was recorded in 155 non-infected and 128 *P. vivax* infected pregnant women.

[n] Definition of gestational outcomes: Abortion–birth < 22nd week of gestation; Stillbirth–fetal death between the 22nd week of gestation and delivery; Preterm birth–birth < 37th week of gestation; Low birth weight–birth weight < 2500 g; Term low birth weight–birth weight < 2500 g from 37th week of gestation; Small for gestational age–birth weight for sex-specific gestational age < 10th centile.

[o] Malaria before current gestation (in all life) and Malaria in previous pregnancy (in other previous pregnancies) were recorded in 158 *P. vivax* infected pregnant women.

**Table 3. Characteristics of the infection during pregnancy.**

| Characteristics | *P. vivax* (N = 159) | *P. vivax* - 1st tri (N = 52) | *P. vivax* - 2nd tri (N = 54) | *P. vivax* - 3rd tri (N = 53) |
|---|---|---|---|---|
| Gestational age at 1st infection, weeks, median (IQR) | 21 (11.0–30.0) | 9.0 (7.0–11.0) | 20.5 (16.0–24.0) | 34.0 (30.0–37.0) |
| Number of infections, n (%) | | | | |
| One infection | 70 (44.0) | 15 (28.8) | 19 (35.2) | 36 (67.9) |
| Two infections | 41 (25.8) | 14 (26.9) | 14 (25.9) | 13 (24.5) |
| Three or more infections | 48 (30.2) | 23 (44.3) | 21 (38.9) | 4 (7.6) |
| Parasitemia at 1st infection, median (IQR) [a] | 997.6 (152.5–3943.0) | 976.1 (226.8–4546.0) | 1019.0 (78.9–3943.0) | 977.6 (157.4–3409.1) |
| Parasitemia 2nd or more infection, median (IQR) | 1052.6 (111.2–4253.0) | 846.8 (83.6–3951.0) | 1132.9 (153.8–3887.0) | 794.1 (113.0–6647.0) |
| Placental malaria, n (%) [b] | | | | |
| No | 117 (91.4) | 34 (91.9) | 40 (90.9) | 43 (91.5) |
| Active Acute | 4 (3.1) | 0 | 2 (4.6) | 2 (4.3) |
| Active Chronic | 0 | 0 | 0 | 0 |
| Past | 7 (5.5) | 3 (8.1) | 2 (4.5) | 2 (4.2) |
| Hemozoin, n (%) [c] | | | | |
| No | 119 (93.7) | 34 (94.4) | 42 (95.5) | 43 (91.5) |
| Mild | 6 (4.7) | 0 | 2 (4.5) | 4 (8.5) |
| Moderate | 2 (1.6) | 2 (5.6) | 0 | 0 |
| Severe | 0 | 0 | 0 | 0 |

Abbreviations: N, total number of individuals; tri, trimester. Results are presented as median and interquartile range (IQR) or total number of events (n) and percentage (%).

[a] Parasitemia at first infection was recorded in 110 *P. vivax* infected pregnant women. Values presented in $10^3$ DNA copies obtained by PET-PCR quantification, after nucleotide extraction from 200 μL of blood cell concentrate.

[b] Placental malaria was assessed in 128 *P. vivax* infected pregnant women.

[c] Hemozoin was assessed in 127 *P. vivax* infected pregnant women. Mild: focal presence in small amounts; moderate: small spots or larger deposits in many locations; severe: large amount presented widely in the tissue sections.

infections closer to term tend to yield higher levels of fibrin deposition and monocytes infiltrate at delivery (**Fig 4B and 4C and S2 Table**). These findings are representative of placental anomalies that can be extended to angiogenic and inflammatory mediators in placental plasma (**S3 and S4 Tables**). First and third trimester infections led to reduced levels of angiopoietin-2 (ANG-2) and increased levels of soluble TEK receptor tyrosine kinase (sTIE-2), respectively (median [IQR], ANG-2 –*Pv*-1st tri 4.4 ng/mL [1.0–6.6] vs. NI 6.8 ng/mL [3.5–12.4], $p = 0.03$; sTIE-2 –*Pv*-3rd tri 18.5 ng/mL [11.7–24.7] vs. NI 13.9 ng/mL [9.1–19.0], $p = 0.008$). Overall, factors such as sFlt-1, sVEGFR-2, and the hormone leptin were also altered in placentas from *Pv*-infected women (**S3 Table**). Regarding inflammatory mediators, only IL-12 and C5a showed more consistent and significant alterations, as they were reduced at delivery due to first and third trimester infections (median [IQR], IL-12 –*Pv*-1st tri 3.3 pg/mL [2.6–3.8] vs. NI 3.6 pg/mL [3.0–4.2], $p = 0.009$; C5a –*Pv*-1st tri 807.1 pg/mL [534.9–913.9] vs. NI 1117.2 pg/mL [814.8–1475.5], $p = 0.03$; *Pv*-3rd tri 673.0 pg/mL [445.9–871.3] vs. NI 1117.2 pg/mL [814.8–1475.5], $p = 0.006$) (**S4 Table**). Altogether, our observations suggest that poor pregnancy outcomes associated with first trimester infections may be linked to placental inflammation, structural abnormalities and angiogenic imbalance during *P. vivax* MiP.

## Antibodies against *P. vivax* MSP1$_{19}$ are not associated with protection from poor pregnancy outcomes

The impact of gravidity on *P. vivax* MiP outcomes is still inconclusive [16,35,36]. Possibly the absence of pregnancy-specific antigens in *P. vivax* and its restricted distribution to areas of low

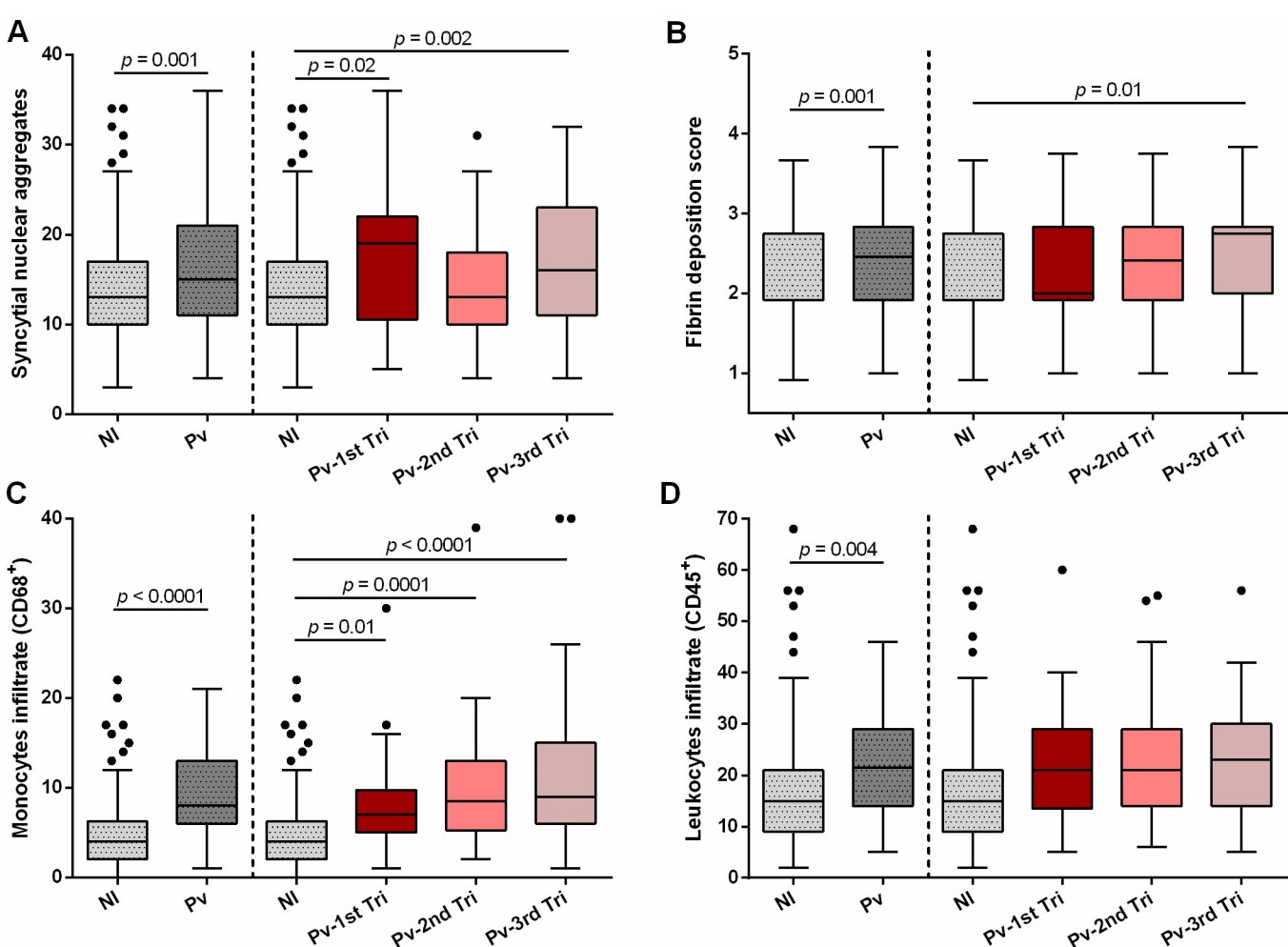

**Fig 4. Evaluation of placental histological parameters according to maternal infection and trimester of first infection.** (A) Syncytial nuclear aggregates; (B) Fibrin deposition score; (C) Monocytes infiltrate (CD68+); (D) Leukocytes infiltrate (CD45+). Histologic parameters were evaluated by microscopy through H&E (syncytial nuclear aggregates and fibrin deposition) or immunohistochemistry (monocytes and leukocytes). NI–non-infected (N = 153–155); Pv–*P. vivax*-infected pregnant women (N = 118–128); Pv-1st tri–*P. vivax* infection 1st trimester (N = 33–37); Pv-2nd tri–*P. vivax* infection 2nd trimester (N = 42–44); Pv-3rd tri–*P. vivax* infection 3rd trimester (N = 43–47). Data are represented as Tukey boxplots, with the bottom and top of each box representing the first and third quartiles; the line inside the box is the median, and the whiskers represent the lowest and the highest data within 1.5 IQR of the first and upper quartiles; and the circles, outliers. Differences between groups were determined by multiple liner regression (adjusted for maternal age, gravidity status, residence, education level, and occupation).

malaria transmission may impair immunity acquisition across pregnancies [8,37]. As such, other antigens might be targeted by the immune system to confer some degree of protection from *P. vivax* MiP. Therefore, we measured antibodies against the C-terminal fragment of the *P. vivax* merozoite surface protein 1 (PvMSP1$_{19}$). We used it as a surrogate marker of previous exposures (**Fig 5A** and **S5 Table**) and sought possible associations between antibody titers and improved pregnancy outcomes. Infected pregnant women who were recruited with more than one malaria episode during their current pregnancy were removed from the analysis, to target antibodies acquired upon any malaria episode occurred before pregnancy. We observed that *Pv*-infected women previously exposed to the parasite (Pv-PvMSP1$^{Ab+}$) had higher titers of IgG when compared with non-infected pregnant women with prior exposure (NI-PvMSP1$^{Ab+}$) (median [IQR], Pv-PvMSP1$^{Ab+}$ 65.1 [14.1–91.6] vs. NI-PvMSP1$^{Ab+}$ 4.0 [2.1–9.8], p < 0.0001) (**Fig 5A** and **S5 Table**). However, no associations were observed between IgG

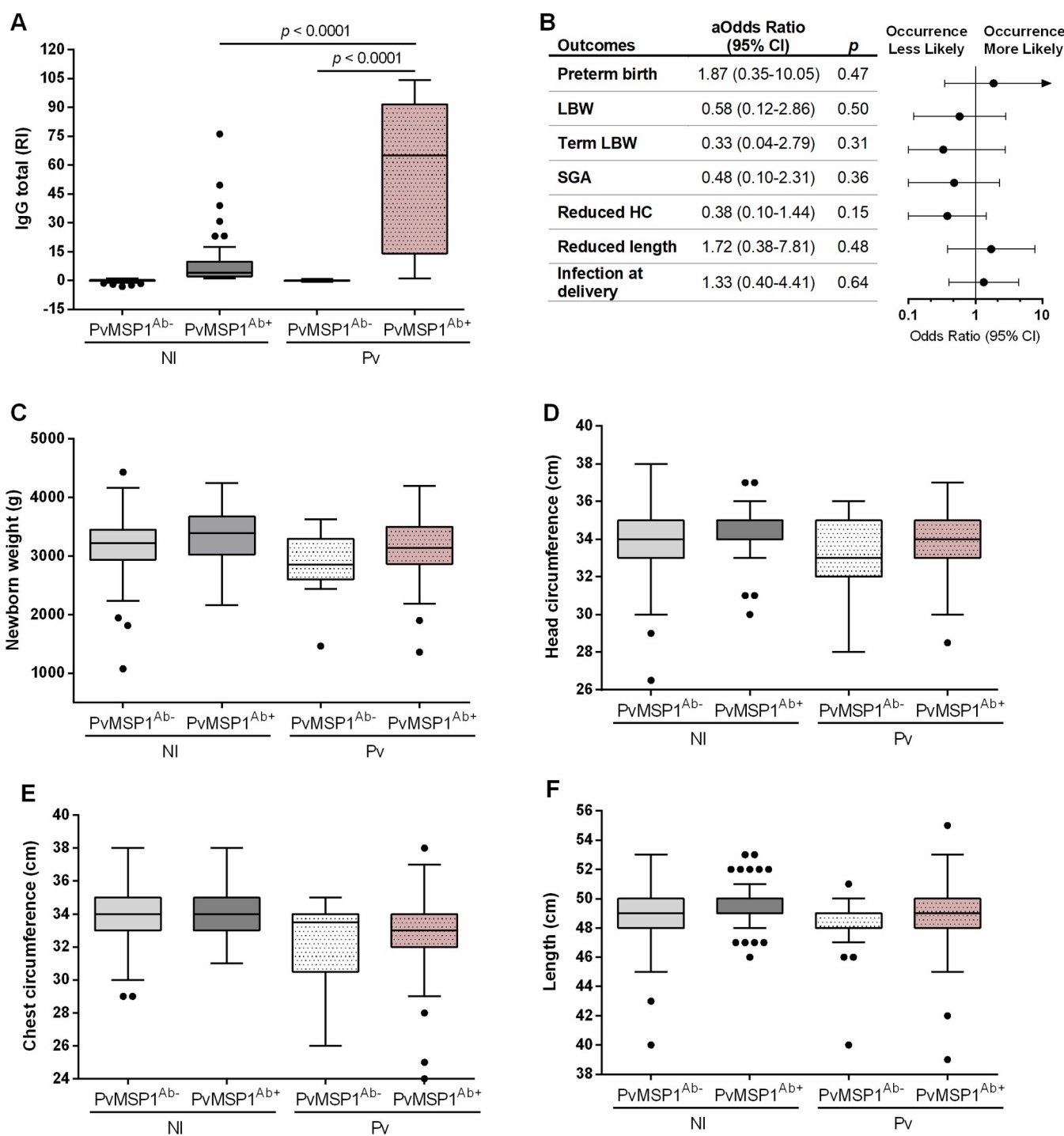

**Fig 5. Anthropometric measurements of newborn, according to previous exposure to *P. vivax* (PvMSP1).** (A) Antibody profile among non-infected and *P. vivax*-infected pregnant women; (B) Forest plot of odds ratios of adverse pregnancy outcomes in infected women with anti-PvMSP1 IgG compared with infected women without anti-PvMSP1 IgG; (C) Newborn weight; (D) Head circumference; (E) Chest circumference; and, (F) Length. NI-PvMSP1$^{Ab-}$—non-infected non-exposed (N = 106–112); NI- PvMSP1$^{Ab+}$—non-infected exposed (N = 54–58); Pv- PvMSP1$^{Ab-}$—*P. vivax*-infected non-exposed (N = 19); Pv-PvMSP1$^{Ab+}$—*P. vivax*-infected exposed (N = 80–95). PvMSP1- *Plasmodium vivax* merozoite surface protein 1; Ab—Antibody; LBW—low birth weight; SGA—small for gestational age; HC—head circumference. RI—Reactivity indices were calculated as the ratio between each test sample's corrected absorbance values and a cut-off value for antigen, corresponding to the average corrected absorbance of samples from 10 malaria-naïve blood donors plus 3 standard deviations. Weight in grams (g). Head and chest circumferences, and body length in centimeters (cm). Data are represented as Tukey boxplots, with the bottom and top of each box representing the first and third quartiles; the line inside the box is the median, and the whiskers represent the lowest and the highest data within 1.5 IQR of the first and upper quartiles; and the circles, outliers. Odds ratios were adjusted (aOR) for maternal age, gravidity status,

residence, education level and occupation using multiple logistic regression. Differences among the groups were determined by multiple linear regression (adjusted for maternal age, gravidity status, residence, education level and occupation).

titers and outcomes, suggesting that the presence of these antibodies do not protect pregnant women from the adverse outcomes of *P. vivax* MiP (**Fig 5B**). Accordingly, no statistically significant differences were observed for birth weight, head and chest circumference, and body length, between *P. vivax*-infected women who had PvMSP1$_{19}$-specific antibodies and those who did not have them (**Fig 5C–5F** and **S5 Table**).

Overall, our results suggest that *P. vivax* infections during the first gestational trimester are associated with poor pregnancy outcomes most likely due to placental histologic and physiologic abnormalities, which are unlikely to be prevented by the presence of IgG against PvMSP1$_{19,}$ acquired during previous *P. vivax*-malaria episodes.

## Discussion

MiP is considered a public health problem worldwide, as it is a significant risk factor for abortion, stillbirth, LBW, and maternal anemia [2–6]. Here, we describe an important association between *P. vivax* MiP and adverse gestational events, placental histopathological changes, and a reduction of the neonate's anthropometric profile. However, a protective association between IgG against PvMSP1$_{19}$ and improved pregnancy outcomes was not observed.

*P. vivax* is widely distributed worldwide, particularly in Asia, Pacific regions, and Latin America. Though, longitudinal studies on gestational *P. vivax*-malaria are still limited, especially regarding placental histopathological analysis [8–11]. Although *P. vivax* MiP is poorly associated with severe malaria, this species' infections are far from benign and may have severe consequences for maternal and fetal health [16,17,35,36,38–40]. The gestational outcomes associated with MiP can vary not only with the species but also with variables such as transmission intensity and stability, number of previous pregnancies, and trimester of infection [7,37]. In our study, no association was found between gravidity and improved pregnancy outcomes (**S6 Table**), suggesting that gestational-dependent mechanisms that govern immunity against *P. falciparum* during pregnancy may not be relevant in infections caused by *P. vivax*. In low transmission settings, where *P. vivax* is more prevalent, gravidity-dependent immunity acquisition is uncertain and inconsistently reported [8,16,35,39]. This could be related to the inexistence of known pregnancy-related *P. vivax* proteins targeted by the immune system [14,15].

The burden of first trimester infections, was scarcely addressed by researchers until recent years [7] despite knowing that perturbation in early pregnancy imposes severe fetal development consequences [41,42]. In our study, *P. vivax* first trimester infections were found associated with LBW and reduced newborn's length. The proportion of pregnancies with LBW observed by us was consistent with other studies conducted in Latin America [38,43] and other *P. vivax* transmission areas [7]. These findings are relevant as LBW is strongly associated with neonatal morbidity and mortality, growth inhibition and cognitive impairment in children, and chronic diseases in adulthood [44]. Similar to what we reported recently on *P. falciparum* MiP, *P. vivax* infections also impact fetal cephalic perimeter development [13]. However, a previous report using the same study population reported that one antenatal *P. vivax* malaria episode in another trimester than the first trimester was enough to impair fetal development, which was aggravated by recurrent parasite exposure during pregnancy [36].

A distinctive characteristic of *P. vivax* is the occurrence of multiple malaria episodes due to reinfections or relapses, with the latter being potentiated by the absence of primaquine treatment in pregnant women due to its teratogenic nature [45]. Taken that and the higher prevalence of peripheral parasitemia observed during the second gestational trimester, also

described in other studies [46,47], we hypothesize that inflammation modulation during pregnancy might favor parasite multiplication and relapses.

The mechanisms by which *P. vivax* infections lead to MiP outcomes are still unclear; yet, systemic events such as maternal anemia and generalized inflammation have been discussed as relevant to *P. vivax* MiP pathology [16,35,40]. Our data evidence a significant hematocrit reduction in pregnant women with *P. vivax*; though, no severe maternal anemia was observed. Only a significant frequency of anemia at delivery was detected among women with third trimester infections. Although maternal anemia is a well-known risk factor for intrauterine growth retardation (IUGR) and LBW in *P. vivax*-infected women [2,16–18,35,36,40], this is unlikely to be influencing pregnancy outcomes in our cohort.

Contrary to *P. falciparum* infections, where pregnancy complications and adverse outcomes are often associated with PM [4,48–50], in *P. vivax* infections, PM is unlikely to be the reason why infected women experience complications during pregnancy. Previous studies indicate that PM caused by *P. vivax* is modest and not associated with placental sequestration of infected erythrocytes [9–11,40], despite evidence of binding to CSA and placental tissue sections [14,15]. Our observations corroborate previous studies, as only 8.6% of the analyzed placentas were classified as having PM. Nonetheless, most placentas from *P. vivax*-infected pregnant women present histopathological changes. Placental lesions and histopathological changes are often difficult to observe in *P. vivax*-infected pregnant women, probably due to the low sample size reported in the studies [9–11,40]. By performing histologic analysis of a considerable number of placentas, we have shown that *P. vivax* infections during the first trimester promote the increase of SNA and monocyte infiltrate at delivery. Monocyte infiltrate and inflammation have been frequently associated with reduced birth weight and preterm birth in pregnant women infected with *P. falciparum* [51–54]. Therefore, we hypothesize that this may also occur in *P. vivax* infections and that placental inflammation independent of PM may be linked to *P. vivax* MiP adverse outcomes, opposing previous observations suggesting that *P. vivax* monoinfections do not promote placental inflammation and that adverse outcomes have an alternative etiologic nature [55].

Dysregulation of angiogenic and inflammatory factors are often associated with placental insufficiency and impaired fetal development in *P. falciparum* MiP [2]. The imbalance of angiopoietins has been associated with PM and LBW in pregnant women infected with *P. falciparum* [56] and observed in *P. vivax*-infected patients with thrombocytopenia [57]. In fact, in *P. vivax* infections, we observed diminished placental levels of ANG-2, sFlt-1, and leptin and increased sTIE-2 and sVEGFR-2 levels at delivery. It is unclear whether placental vasomodulation is being promoted or inhibited in our cohort, as diminished sFlt-1 and increased sVEGFR-2 support angiogenesis and vasodilation [58], while decreased ANG-2 and increased sTIE-2 suggest an anti-angiogenic environment [59]. On the other hand, in our study, the placental levels of classic inflammatory cytokines at delivery were not altered, reflecting a picture from past events. It is plausible that cytokines at delivery are already downregulated, which is in line with previous observations that showed inflammatory cytokines associated with episodes of *P. vivax* malaria during pregnancy that are lost at delivery [60]. Yet, a decrease in the plasma levels of IL-12 and C5a at delivery, particularly upon infections that occur during the first and third trimesters, was observed by us. IL-12 has been previously associated with *P. vivax* infections during pregnancy [60], and C5a is considered a hypothetical key driver of PM pathogenesis, being associated with placental insufficiency and SGA babies, and correlated negatively and positively with ANG-1 and ANG-2, respectively [61,62]. Moreover, our study shows that both placental C5a and ANG-2 are diminished at delivery when women are infected for the first time during the first trimester. These observations oppose a previous hypothesis assuming that *P. vivax* infections would promote the same augmented levels of C5a

as those observed during *P. falciparum* MiP [61]. Altogether, our data suggest that inflammatory responses acting against *P. vivax* infections in the first trimester may contribute to chronic placental damage and insufficiency. This might occur through angiogenic and vascular remodeling mechanisms, which, when altered, promote abnormal placental/fetal development and poor pregnancy outcomes, despite early pharmacological intervention with antimalarials.

Currently, the main strategy to control/treat *Plasmodium* spp. infections relies on antimalarials and vaccines, channeling the research resources to develop these approaches. Regarding *P. falciparum* MiP, vaccines such as PAMVAC have been tested to induce immunity acquisition to VAR2CSA. However, there are no known orthologs in *P. vivax* that the immune system can directly target to protect women during pregnancy. Alternatively, the combat of disease symptoms in pregnant women can be done through the natural generation of antibodies against blood-stage parasite antigens. Antibodies against *P. vivax* MSP1$_{19}$ were shown to be associated with protection against symptomatic malaria in non-pregnant individuals [63,64] and those against *P. vivax* VIR, and Duffy-binding protein (PvDBP) were shown protective of birth weight reduction in pregnant women infected with *P. vivax* [65,66], suggesting that targeting non-pregnancy related antigens may also improve gestational outcomes. Nonetheless, in our study, we did not observe a statistically significant association between antibodies against *P. vivax* MSP1$_{19}$ and improved pregnancy outcomes, which is in line with previous observations showing that *P. vivax* MSP1$_{19}$ antibodies are associated with infection but do not correlate with protection from maternal anemia or reduced birth weight [66].

*P. vivax* MiP is still poorly understood, representing major challenges for pregnant women and their concepts. Here, we have shown that *P. vivax* MiP impairs newborn's development and leads to placental tissue damage. We conclude that *P. vivax* infections during the first trimester are associated with poor pregnancy outcomes and that the presence of antibodies against *P. vivax* MSP1$_{19}$ does not protect pregnant women from adverse gestational outcomes. Abnormal pregnancies in this context might occur due to placental inflammation and dysregulation of local homeostasis, regardless the absence of parasite sequestration and prompt treatment administration given to pregnant women in the endemic areas of the Brazilian Amazon.

## Supporting information

**S1 STROBE checklist.**
(DOCX)

**S1 Table. Characteristics of Anthropometric Measurements of Newborns, according to the gestational trimester in which the first infection occurred.**
(DOCX)

**S2 Table. Placental Parameters of Non-infected and *P. vivax*-infected women, according to the gestational trimester in which the first infection occurred.**
(DOCX)

**S3 Table. Angiogenic factors in placental plasma from Non-infected and *P. vivax*-infected women, according to the gestational trimester in which the first infection occurred.**
(DOCX)

**S4 Table. Inflammatory factors in placental plasma from Non-infected and *P. vivax*-infected women, according to the gestational trimester in which the first infection occurred.**
(DOCX)

**S5 Table. Anthropometric characteristics of the newborns, according to maternal previous exposure to *P. vivax* (PvMSP1).**
(DOCX)

**S6 Table. Association between adverse pregnancy outcomes and *P. vivax* infections according to gravidity.**
(DOCX)

**S1 Text. Analysis plan.**
(DOCX)

**S1 Data. Study database.**
(XLSX)

## Acknowledgments

We thank the pregnant women from "Vale do Juruá" who agreed to participate in the study, as well as the teams from the Hospital da Mulher e da Criança do Juruá and Gerência de Endemias/SESACRE for their invaluable assistance. Additionally, we thank the Universidade Federal do Acre, and the direction of Santa Casa de Misericórdia de Cruzeiro do Sul for the support.

## Author Contributions

**Conceptualization:** Jamille Gregório Dombrowski, Claudio Romero Farias Marinho.

**Data curation:** Jamille Gregório Dombrowski, Claudio Romero Farias Marinho.

**Formal analysis:** Jamille Gregório Dombrowski, André Barateiro, André Boler Cláudio da Silva Barros.

**Funding acquisition:** Carsten Wrenger, Giuseppe Palmisano, Sabrina Epiphanio, Claudio Romero Farias Marinho.

**Investigation:** Jamille Gregório Dombrowski, André Barateiro, Erika Paula Machado Peixoto, Rodrigo Medeiros de Souza, Taane Gregory Clark, Susana Campino, Gerhard Wunderlich, Lígia Antunes Gonçalves, Claudio Romero Farias Marinho.

**Methodology:** Jamille Gregório Dombrowski, André Barateiro, Erika Paula Machado Peixoto, Taane Gregory Clark, Susana Campino, Gerhard Wunderlich, Lígia Antunes Gonçalves, Claudio Romero Farias Marinho.

**Project administration:** Jamille Gregório Dombrowski, Claudio Romero Farias Marinho.

**Resources:** Carsten Wrenger, Gerhard Wunderlich, Claudio Romero Farias Marinho.

**Supervision:** Jamille Gregório Dombrowski, Claudio Romero Farias Marinho.

**Validation:** Jamille Gregório Dombrowski, Taane Gregory Clark, Claudio Romero Farias Marinho.

**Writing – original draft:** Jamille Gregório Dombrowski, André Barateiro.

**Writing – review & editing:** Jamille Gregório Dombrowski, André Barateiro, Erika Paula Machado Peixoto, André Boler Cláudio da Silva Barros, Rodrigo Medeiros de Souza, Taane Gregory Clark, Susana Campino, Carsten Wrenger, Gerhard Wunderlich, Giuseppe Palmisano, Sabrina Epiphanio, Lígia Antunes Gonçalves, Claudio Romero Farias Marinho.

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
