## [Decision Letter · Decision Letter 0]

11 Dec 2020

Dear Dr. Marinho,

Thank you very much for submitting your manuscript "Adverse pregnancy outcomes are associated with P. vivax malaria in a prospective cohort of women from the Brazilian Amazon" for consideration at PLOS Neglected Tropical Diseases. As with all papers reviewed by the journal, your manuscript was reviewed by members of the editorial board and by several independent reviewers. In light of the reviews (below this email), we would like to invite the resubmission of a significantly-revised version that takes into account the reviewers' comments. 

We cannot make any decision about publication until we have seen the revised manuscript and your response to the reviewers' comments. Your revised manuscript is also likely to be sent to reviewers for further evaluation.

Sincerely,

Hans-Peter Fuehrer

Deputy Editor

Hans-Peter Fuehrer

Deputy Editor

Reviewer's Responses to Questions

**Key Review Criteria Required for Acceptance?**

**Methods**

-Are the objectives of the study clearly articulated with a clear testable hypothesis stated?

-Is the study design appropriate to address the stated objectives?

-Is the population clearly described and appropriate for the hypothesis being tested?

-Is the sample size sufficient to ensure adequate power to address the hypothesis being tested?

-Were correct statistical analysis used to support conclusions?

-Are there concerns about ethical or regulatory requirements being met?

Reviewer #1: The authors have studied the impact of P. vivax infection on pregnancy outcome in a large prospective cohort of Brazilian women. Maternal history of infection during the pregnancy, placental histopathology, levels of angiogenic factors and cytokines, and levels of PvMSP1-19 antibodies are related to various measures of pregnancy outcome. The authors conclude that P. vivax infection, particularly during the first trimester, negatively affects pregnancy outcome.

The authors use the terms malaria in pregnancy and placental malaria more or less randomly. In my opinion, placental malaria should be reserved for infections, where a placental disease focus is known or highly likely. Whereas this is generally the case for P. falciparum infection among women living in areas with stable parasite transmission, it remains highly controversial whether selective placental accumulation of P. vivax-infected erythrocytes occurs. As the authors do not provide such evidence, I strongly recommend that the authors use the term malaria in pregnancy (or pregnancy-associated malaria) throughout. This recommendation is supported by the fact that the authors did not find a statistically significant relation between their outcome variables and parity. Such a relation would be strongly expected in the case of bona fide placental malaria.

The study appears well designed and it is nicely presented. My overriding concern is whether the uninfected and infected women can be compared directly, as they are throughout the manuscript. The reason for this concern is Table 1, where it is stated that 44/159 infected women lived in a rural area, whereas only 8/170 of the non-infected did. This difference is very highly significant – but not commented on in the manuscript. It is therefore an open – and critical – question whether it is justified to attribute the differences between the infected and uninfected women (only) to P. vivax infection. While it appears reasonable to suspect that a strongly pro-inflammatory infection such as P. vivax malaria would negatively affect pregnancy, it is also very highly likely that rural residence for many reasons other than P. vivax exposure predisposes for a pro-inflammatory status. The authors should comment on this, and definitely control for this likely confounding factor in their analyses wherever possible (i.e., a multivariate approach). Additional factors reported in Table 1 that might bias the data in the same direction (albeit less pronounced manner) are age, occupation, and educational level.

Reviewer #2: Objectives of this study are well stated. Many important details of the methods, especially of sampling from the participants, are omitted. I expect that previous publications provide this information, but readers of this article should not have to seek out those papers in order to understand the study. 

It would be helpful to know (briefly) how the women in this study were recruited – before or during pregnancy, at home or clinic or hospital, and how followed – monthly, active or passive follow-up? What exactly defines a woman ‘with P vivax?’ – detected infection at any time in the pregnancy, or at parturition, or in the placenta, or any of these? In general we need to be told when samples were taken – some can obviously be taken only at or around parturition, but others could be taken at various times during pregnancy: e.g. at what point in the study were blood samples taken from women studied for total IgG antibodies against Plasmodium vivax Merozoite Surface Protein 1 (PvMSP1-19)? (Was evidence being sought for P vivax infection before pregnancy had started, or at some stage in the current pregnancy before enrolment?)

Were tests for Pv taken at first recruitment irrespective of symptoms (e.g. fever); were women asked to attend for any fever; were blood films taken routinely at scheduled visits to the clinic, or only if there were symptoms? Did the ‘endemic surveillance team’ actively seek women with fever, or women in pregnancy, to enrol into the study? 

Page 11. When assessed by microscopy, were placental samples ‘blinded’ as to P vivax infection in the mother?

Reviewer #3: - Details on the timing of recruitment and follow up in the study are not adequately described. At what gestational age were women recruited into the study? How many malaria episodes occurred during pregnancy?

- Given that the women were not followed during pregnancy unless they were symptomatic, how can the authors rule out that they did not experience a P. falciparum infection at some point in pregnancy that was asymptomatic? 

- Table 1 should include statistical analyses to compare across the different groups.

- How were previous P. vivax infections in the current or prior pregnancies confirmed? (Line 312)

**Results**

-Does the analysis presented match the analysis plan?

-Are the results clearly and completely presented?

-Are the figures (Tables, Images) of sufficient quality for clarity?

Reviewer #1: Univariate analyses are used throughout. A multivariate approach is strongly recommended (see preceding section).

Reviewer #2: Many data are presented in detail in this paper, both in text and Figures, so that the important findings are difficult to identify. It would be useful to decide a hierarchy of importance of findings and make sure that these are conveyed prominently.

P14 line 290: “All p values were 2-sided, and interpreted at a significance level of 0.05.” Mention should be made that there are very large numbers of statistical tests in the Tables and figures – will any adjustment for multiple testing be made to the interpretations of ‘significance’? 

In Fig 4 please clarify why denominators in column 2 don’t add up to the same total on each line.

Reviewer #3: - In Fig 8, why would newborns have longer length in the uninfected group depending on their PvMSP1 Ab status?

- It seems possible that the PvMSP1 antibodies at enrolment could be due to a recent infection in that pregnancy prior to enrolment, rather than an exposure prior to pregnancy. Consideration of the timing of infection and timing of enrolment are important to interpret these serology data.

**Conclusions**

-Are the conclusions supported by the data presented?

-Are the limitations of analysis clearly described?

-Do the authors discuss how these data can be helpful to advance our understanding of the topic under study?

-Is public health relevance addressed?

Reviewer #1: Until the data have been re-analysed with inclusion of confounding factors, I cannot confidently evaluate the appropriateness of the authors’ interpretations of their data, or the conclusions they draw.

Reviewer #2: There is frequent reference (or implication) in this paper to a finding being the cause of some later event or difference. For example: p32 lines 557-558: “we have observed an important impact of P. vivax infection on fetal growth…”. In fact association has been observed, but you have not demonstrated a causal link (implied by ‘impact’). The above, for example, would be better written as ‘we have observed a significant association between P vivax infection and impaired fetal growth…’. Several implications or assumptions of causality recur throughout the paper and would be better reworded appropriately (including in the last sentence of Discussion).

Reviewer #3: - The statement on lines 73-75 is not entirely accurate. The study by Bardaji et al (2017), for example, explored some of these interactions between Pv infection in pregnancy and birth outcomes.

- The statement on lines 83-85 should be revised since these associations were observed in specific groups, not generally between infected vs non-infected women, except for preterm birth as shown in Fig 2. The statement should also reflect the data in Figures 3 and 4. 

- Line 86. Placental malaria is defined by the presence of parasites or hemozoin in the placenta, not by these histological changes in the placenta. This sentence should be revised accordingly. Only 8 women in the study had placental malaria and the results should be interpreted with some caution. 

- I don’t agree with the statement on line 351. The difference is only significant for infections in the second trimester. The results seem to be skewed by the high error bars.

- The strong conclusions from the serology data for PvMSP1-19 are premature. The authors state that they used this marker as a surrogate of past exposure to P. vivax and yet they conclude that these antibodies are associated with protection. Was there a correlation between PvMSP1 Abs and placental outcomes or clinical outcomes? The discussion of pre-existing immunity should be removed or revised substantially.

**Editorial and Data Presentation Modifications?**

Reviewer #1: (No Response)

Reviewer #2: One woman in her first pregnancy is a primigravida, more than one are primigravidae. Please correct ‘primigravida’ (and ‘multigravida’) throughout the paper, including in Tables and Figures, where they should be plural nouns. p.21 line 348: 'Pv-infected women experienced more adverse pregnancy outcomes'…Several references are made to ‘more’ or ‘higher’ – when referring to differences between groups; in these situations please add ‘on average’ or ‘mean’ or ‘median’ before the comparative word, and make sure you indicate with whom or against what the comparison is being made. For example P32 line 571 “we observed that primigravida who presented infection in the second trimester had higher parasitemia.” Higher than whose? Than you would expect? Or than multigravidae with 2nd trimester infections? Or than primigravidae with infections in the first trimester? Etc… Please look out for comparatives without a comparator, and clarify accordingly.

Throughout the paper, especially in the Discussion, redundant words are commonly used to start sentences. Try omitting these and see if it makes any difference to the clarity of the text (I don’t think it does): examples:

‘Additionally’, ‘Finally’ (best avoided because you can never say the final word on any subject), ‘Not surprisingly’, ‘Further’, ‘In addition’, ‘Remarkably’, ‘Furthermore’, ‘Moreover’, ‘Controversially’, ‘Thus’.

Reviewer #3: Minor comments

- Line 159. Please state EIR or some other measure of malaria transmission in the study area. 

- Specify source of PvMSP1-19 protein used for ELISAs and concentration coated on the plate.

- Table 3. Include the units for the parasitemia 

- Minor grammatical errors, as follows:

- Line 61, delete ‘occurring’

- Line 122, rephrase ‘it is detected’

- Line 150, change the term ‘benignity paradigm’

- Line 183, should be ‘first’

- Line 240 should be ‘10th percentile’

- Line 308 should be ‘pregnant women’

- Line 316, change ‘exert’ to ‘may be detrimental’

- Line 377, remove ‘the’

- Line 379, ‘in’ should be ‘on’

- Line 385, Change the word ‘patent’, perhaps to ‘apparent’ or another term

- Line 387, change ‘multigravidity status’ to ‘in multigravida’

- Line 430, should be ‘alterations in several’

- Line 457 should be ‘cytokine’

- Line 460, remove period before bracket

- Line 461, change ‘of’ to ‘in’

- Line 484, please change the term ‘tenuous’ – the meaning here is not clear

- Line 489, change ‘heighten’ to ‘greater’

- Line 509, change to samples collected at enrollment

- Line 540 is repetitive with line 535

- Line 560, ‘of’ should be ‘on’

- Line 561, add a comma after infections

- Line 579, remove ‘the existence of’, line 580 ‘supports’

- Line 602, change ‘though’ to ‘however’

- Line 627 should be ‘protection’

**Summary and General Comments**

Reviewer #1: (No Response)

Reviewer #2: The importance of this study is diminished by the profusion of material - more brevity in the introduction and discussion, and a clarification of findings of importance, would make it more accessible to readers. At present the paper would do better for a thesis-Chapter than a journal publication.

Reviewer #3: The authors present the results of a longitudinal study of P. vivax malaria in a population of pregnant women in Brazil. This study focuses exclusively on women with P. vivax infection identified through passive case detection and at delivery compared to non-infected pregnant women. Extensive data are provided on the clinical outcomes, placental histology, newborn characteristics and analyses of various biomarkers. The study reveals that P. vivax infection in pregnancy, particularly during the first trimester, can be detrimental to the health of the newborn, prompting re-consideration of the severity of infections with this species in pregnancy. While the study provides very important data on P. vivax MiP that will contribute significantly to the field, there are a number of limitations and these should be discussed.

PLOS authors have the option to publish the peer review history of their article (what does this mean?). If published, this will include your full peer review and any attached files.

Reviewer #1: No

Reviewer #2: No

Reviewer #3: No
---

## [Decision Letter · Decision Letter 1]

15 Apr 2021

Dear Dr. Marinho,

We are pleased to inform you that your manuscript 'Adverse pregnancy outcomes are associated with P. vivax malaria in a prospective cohort of women from the Brazilian Amazon' has been provisionally accepted for publication in PLOS Neglected Tropical Diseases.

Best regards,

Hans-Peter Fuehrer

Deputy Editor

Editor: Write out Plasmodium in title

Reviewer's Responses to Questions

**Key Review Criteria Required for Acceptance?**

**Methods**

-Are the objectives of the study clearly articulated with a clear testable hypothesis stated?

-Is the study design appropriate to address the stated objectives?

-Is the population clearly described and appropriate for the hypothesis being tested?

-Is the sample size sufficient to ensure adequate power to address the hypothesis being tested?

-Were correct statistical analysis used to support conclusions?

-Are there concerns about ethical or regulatory requirements being met?

Reviewer #1: Adequate methodology with no remaining concerns.

Reviewer #3: The revisions are acceptable.

**Results**

-Does the analysis presented match the analysis plan?

-Are the results clearly and completely presented?

-Are the figures (Tables, Images) of sufficient quality for clarity?

Reviewer #1: Results adequately presented and analysed.

Reviewer #3: The revisions are acceptable.

**Conclusions**

-Are the conclusions supported by the data presented?

-Are the limitations of analysis clearly described?

-Do the authors discuss how these data can be helpful to advance our understanding of the topic under study?

-Is public health relevance addressed?

Reviewer #1: Appropriate conclusions justified by the results obtained.

Reviewer #3: The revisions are acceptable.

**Editorial and Data Presentation Modifications?**

Reviewer #1: None.

Reviewer #3: (No Response)

**Summary and General Comments**

Reviewer #1: All my concerns regarding the earlier version of this manuscript have been alleviated. I congratulate the authors with a nice study that constitutes a valuable addition to the litterature.

Reviewer #3: The authors have substantially revised the manuscript based on their new statistical analyses. All concerns that I raised have been addressed in this new version.

PLOS authors have the option to publish the peer review history of their article (what does this mean?). If published, this will include your full peer review and any attached files.

Reviewer #1: **Yes: **Lars Hviid

Reviewer #3: No

---

## [Editor Report · Acceptance letter]

22 Apr 2021

Dear Dr. Marinho,

We are delighted to inform you that your manuscript, "Adverse pregnancy outcomes are associated with P. vivax malaria in a prospective cohort of women from the Brazilian Amazon," has been formally accepted for publication in PLOS Neglected Tropical Diseases.

Best regards,

Shaden Kamhawi

co-Editor-in-Chief

Paul Brindley

co-Editor-in-Chief
